# An Update on Complications Associated with SARS-CoV-2 Infection and COVID-19 Vaccination

**DOI:** 10.3390/vaccines10101639

**Published:** 2022-09-29

**Authors:** Purvita Chowdhury, Shinjini Bhattacharya, Bhaskarjyoti Gogoi, Ravindra P. Veeranna, Sachin Kumar

**Affiliations:** 1Department of Biosciences and Bioengineering, Indian Institute of Technology Guwahati, Guwahati 781039, India; 2Department of Biotechnology, The Assam Royal Global University, Guwahati 781035, India; 3Department of Biochemistry, CSIR-CFTRI, KRS Road, Mysuru 570020, India

**Keywords:** COVID, vaccines, coronavirus, pathogenesis, immunity

## Abstract

Over the past two years, SARS-CoV-2 has dramatically spread worldwide and emerged as a major pandemic which has left an unprecedented mark on healthcare systems and economies worldwide. As our understanding of the virus and its epidemiology continues to grow, the acute phase clinical symptoms and long-term and vaccine-related complications are becoming more apparent. With heterogeneity in presentations, comparisons may be drawn between COVID-19-related sequelae and vaccination related adverse events. The present review article aims to address the currently available literature on the SARS-CoV-2 virus, immune responses, the pathophysiology of clinical presentations, and available vaccinations with its adverse events for the appraisal of its potential impact on the COVID-19 management system.

## 1. Introduction

The initial outbreak of the unknown viral pneumonia-like disease in Wuhan, China, in early December 2019, rapidly spread across the globe within a short period. Nucleotide sequencing of the virus revealed it to be a novel strain of coronavirus belonging to the genus Beta coronavirus and was named Severe Acute Respiratory Syndrome Coronavirus 2 (SARS-CoV-2) [1]. By the end of January 2020, over ten countries from Asia, Europe, North America, and Australia were reporting cases of rising person-to-person transmissibility. The World Health Organization (WHO) soon declared a global health emergency on 31 January and a pandemic by 11 March 2020 [2,3]. As per the WHO data, 532,201,219 cases with more than 6 million mortality (until June 2022) have been recorded globally, making it the most significant global health crisis since the 1918 influenza pandemic. During the pandemic, several variants of SARS-CoV-2 emerged and spread rapidly with a few potent strains globally, and few of them have been responsible for the recurrent waves of the disease [4]. The global efforts to control and manage the pandemic were initially met with setback because of the highly transmissible and elusive nature of the virus-like alpha, beta, gamma, delta, and omicron variants that were subsequently identified as “variants of concern” [4]. However, a concerted approach to the pandemic resulted in the development of novel vaccines against COVID-19. By June 2022, the WHO reported 166 vaccine candidates under clinical phase development, out of which 33% are protein subunit vaccine candidates [5]. Different platforms for vaccine development have been explored and utilized by the researchers, including protein subunits, replicating and non-replicating viral vectors, DNA and RNA, inactivated viruses, virus-like particles, live attenuated viruses, as well as a bacterial antigen-spore expression vector. It is worth mentioning that vaccines’ ability to induce effectiveness, T cell responses, and neutralizing antibodies were observed to be different in respect to the different variants of the virus [6]. In December 2020, the US Food and Drug Administration (FDA), after rigorous review, allowed Pfizer-BioNTech-developed mRNA vaccine (BNT162) for emergency use authorizing it in the USA for the administration of mass vaccination [7]. This was the first instance of administrating mass COVID-19 vaccination in any country. As of June 2022, more than 11 billion doses of various vaccines against COVID-19 have been administered worldwide [8].

Therefore, it is pertinent to understand the host immune response along with the clinical spectrum of COVID-19 and its long-term sequelae. The current review aims to explore and apprise COVID-19 infections and their vaccination related complications.

## 2. The Virus and Its Effect

Among the beta coronaviruses that have infected humans in the past decades, such as SARS-CoV and the Middle East respiratory syndrome (MERS-CoV), SARS-CoV-2 has had the most devastating effect. Although these coronaviruses share 79.6% sequence similarity, variations in the epidemiology of these viruses are evident [9,10]. Since the outbreak of the disease, human-to-human transmission of the SARS-CoV-2 has been established. However, the specific route of natural transmission to humans is dubious. It has been speculated as a spillover event from bat SARS-related coronavirus (RaTG13), which shares more than 96% whole genome sequence similarity [11]. Being one of the most significant single stranded positive sense RNA viruses, the genomic size of the SARS-CoV-2 is around 30 kilobases and comprises 12 protein-coding genes. The genes encode 16 non-structural proteins-Nsp1-16 required for viral replication and assembly, four structural proteins-Spike (S), envelope (E), membrane (M) and nucleocapsid (N) essential for viral entry; and five accessory proteins-ORF3, ORF4a, ORF4b, ORF 5, ORF 8 that are unique to SARS-CoV-2 [12,13]. SARS-CoV-2 has a stronger affinity to the host receptor, i.e., Angiotensin converting enzyme 2 (ACE 2), than other human infecting coronaviruses and has an additional cleavage site, which allows higher infectivity and more severe longer-term complications [14].

### Variants of SARS-CoV-2

Since its emergence, the SARS-CoV-2 virus has evolved dramatically from its ancestor wild type with increasing transmissibility and virulence. This natural selection is based on mutations common for RNA viruses, which are beneficial in terms of replication, host immune evasion mechanism, and transmission of the virus [15]. Thus, SARS-CoV-2 with mutations are defined as variants, while a “variant of concern” (VOC) signifies a variant with increased transmissibility and risk of severity, a significant decrease in neutralizing antibodies produced in response to vaccination/previous infection, or decreased efficiency of vaccines/treatment [16]. The SARS-CoV-2 VOC are described in Table 1.

## 3. Effect of SARS-CoV-2 Infection on Host Immunity

The human population depicts a very diverse range of responses to COVID-19 infections. It has been suggested that the reason for the disease’s variability lies in the host’s genetic makeup, mostly in the immune-related genes [20]. The innate immune response to viral infections is activated within hours of viral exposure as the first line of defence releasing antiviral molecules at the infection site. This involves the Pattern Recognition Receptors (PRRs) such as TLR7 and TLR8 in case of single-stranded RNA viruses, RIG-I-like (RLR), NLR expressed by epithelial cells, and local immune cells such as alveolar macrophages. These PRRs recruit other adaptor proteins on ligand binding, activating downstream transcription factors such as interferon regulatory factor (IRF), NF-κB and AP-1 that produce different chemokines and Type-I and Type-III antiviral interferons [21]. Reports suggest SARS-CoV-2 infection can also be detected through the cytosolic DNA sensing cyclic GMP-AMP synthase (cGAS)—stimulator of interferon genes (STING) pathway [22]. Various transcription factors are activated as a result of viral detection, which further causes the secretion of pro-inflammatory chemokines and cytokines, e.g., tumour necrosis factor-alpha (TNF-α), interleukin (IL)-1, IL-6 by macrophages, monocytes, dendritic cells (DC), neutrophils [23]. Early release of cytokines and adequate production and proper localization of effector cells often controls COVID-19 infection successfully but patients with severe infection suffer from cytokine release syndrome (CRS), also known as cytokine storm, characterized by dampened type I interferon response and increased level of antiviral cytokines [24,25]. SARS-CoV-2 manipulates the host antiviral immune response and enhances the virus entry by making the type I interferon responses highly inefficient [26].

The adaptive immune response initiated a few days post viral infection, requires priming by the components of the innate immune system. DC, monocytes, and macrophages maturation into antigen-presenting cells (APCs) depend on the induction by type I interferon responses. The APCs play a major role in activating naïve clusters of differentiation (CD) 4^+^, CD8^+^ T-lymphocytes, and regulatory T lymphocytes (Treg) [27,28]. In recuperating COVID-19 patients, a pattern of antigen immunodominance has been observed where nine viral proteins are shown to be responsible for 83% of total CD4^+^ T lymphocyte response and eight viral proteins are responsible for 81% of total CD8^+^ T lymphocyte response [29]. Patients suffering from severe COVID-19 diseases have reduced APCs, and natural killer (NK) cells, hence having reduced antigen presenting potential [30,31]. The neutrophil to lymphocyte ratio, characteristic of inflammation and depleted CD4^+^ T lymphocytes has been observed and it correlates to the disease severity [32]. Humoral immune response, mediated by antibodies in SARS-CoV-2 infection is elicited against the N and S proteins of the virus [33,34]. SARS-CoV-2 neutralizing antibodies such as IgA, IgM and IgG have been recognized in COVID-19 patients [34,35]. The interaction of these antibodies with NK cells via CD16 Fc receptor binding induces antibody-dependent cell-mediated cytotoxicity. These cytotoxic responses are further induced by CD8^+^ T lymphocyte via release of soluble factors, e.g., perforin and granzymes production, which are increased in patients with severe COVID-19 disease [27,28,36]. Several facets of B-cell and T-cell immune responses to different stages of COVID-19 disease remain unclear and further investigation may provide more understanding about the virus-host interaction that will facilitate the development of therapeutics against the disease [37].

## 4. Clinical Spectrum

The clinical manifestations of COVID-19 range from asymptomatic to severe illness and fatality. The National Institutes of Health (NIH) characterized COVID-19 into five categories: asymptomatic, mild, moderate, severe, and critical illness, based on clinical, laboratory, and radiographic diagnosis as well as organ functions [38].

Interestingly, not everyone infected with the virus develops the characteristic phenotypical presentations. People with a positive laboratory diagnoses for SARS-CoV-2 and without any clinical manifestations are termed asymptomatic infections [38]. A recent meta-analysis and review that included around 30,00,000 individuals suggested that globally 0.25% (95% CI, 0.23–0.27%) of the population and 40.50% (95% CI, 33.50–47.50%) of the COVID-19 confirmed population are asymptomatic infections [39]. An increased proportion of asymptomatic infections has also been reported among pregnant women hospitalized for childbirth [40,41]. A preliminary review also accounted for 40–45% of the COVID-19 confirmed population to be asymptomatic [42]. Although not fully understood, multiple factors such as the immune response, genetic resistance/susceptibility, virulence of the virus strain and extent of exposure could be attributed to such differences in symptoms.

Mild infections of SARS-CoV-2 include upper respiratory tract symptoms without dyspnea, abnormal chest imaging and shortness of breath [43]. While individuals with severe infections present lower respiratory tract symptoms, oxygen saturation levels <94% of room air and respiratory rate of >30 breaths per minute. During the initial period of the pandemic, the Chinese Centers for Disease Control and Prevention (CDC) group conducted a study including more than 44,500 confirmed cases and observed that 81% (*n* = 36,160) of the cases had mild symptoms. Approximately 14% (*n* = 6168) showed severe symptoms with 2.3% (*n* = 1023) case fatality rate. Notably, cases above 80 years of age group had a higher case fatality rate of 14.8% compared to other age groups [44]. Another cross-sectional study from the United States reported an 11.4% fatality rate of hospitalized adults from March to December 2020 [45]. However, this proportion can be misleading, especially in low-income countries with limited medical facilities. An observational study comprising ten African countries recorded 48.2% (95% CI 46.4–50.0) in-hospital mortality within a span of 30 days [46]. It is noteworthy that risk factors of severe clinical symptoms have also been found to differ according to the virus variants, age, sex, socio-economic background, comorbidities, and genetic factors of the patients.
**(i)** **Age and sex**: SARS-CoV-2 infection has been documented across all age groups; however, individuals in the 40–80 age group seem to be at higher risk of disease severity. During the first pandemic wave, a hospitalized confirmed Chinese cohort comprising 36,358 patients was monitored for the risk factors for COVID-19 related fatalities. The majority of the patients belonged to the middle age group and increased age was significantly an independent risk factor for death (OR = 1.061 [95% CI 1.057–1.065], *p* < 0.001) [47]. A meta-analysis of the isolated effect of age found an increased risk of case mortality and hospitalization by 7.4% and 3.4%, respectively, with the increase in age years [48]. Another population cohort study with more than 450,000 participants reported a 13-fold increased risk of COVID-19 related mortality in >75 years aged individuals (95% CI 9.13–17.85) as compared to lower age groups [49]. In contrast, a 2.7% fatality rate was observed among hospitalized young adults in 18–35 years age group [50].

In many cohorts from across the world, the male gender has been found to be at a higher risk of COVID-19 related severity [51,52,53]. In a Chinese cohort, the male gender was significantly associated with higher odds of mortality from COVID infection (OR = 1.585 [95% CI 1.301–1.933], *p* < 0.001) [47].

**(ii)** **Comorbidities**: With the gradual development of the pandemic, multiple studies have documented comorbidities as risk factors for severity of the disease and COVID-19 related mortality. Some of the risk factors are given in Table 2.

**(iii)** **Genetic factors**: A number of studies exploring Genome-Wide Association Studies (GWAS), COVID-19 Host genetics initiative (HGI), whole exome sequencing (WES), 23 and ME, among others, have underlined the importance of host genetic factors for susceptibility to the severity of the disease [57,58]. Table 3 below represents a few examples of the genetic factors associated with COVID-19 severity.

## 5. Complications during COVID-19

Given a wide range of clinical manifestations due to COVID-19, around 15% of the patients develop severe pneumonia, while approximately 5% exhibit acute respiratory distress syndrome (ARDS), multiple organ failure and septic shock [65,66]. Complications due to acute infections have been reported under respiratory, cardiac, haematological, and neurological symptoms [67]. Usually, these symptoms resolve within four weeks of infection. However, globally there have been reports of persistent and prolonged clinical presentations, including pulmonary, cardiovascular, haematological, neuropsychiatric and autoimmune in the post-acute phase of COVID-19 [68,69] (Figure 1).

The pathophysiology of post COVID-19 complications is multifactorial and it is mostly speculated that SARS-CoV-2 infects a diverse range of human cell types making it a highly efficient infection. In fact, the virus receptor ACE2 is expressed in various tissues, such as the respiratory system, blood vessels, cardiomyocytes, brain endothelium and renal tissues among others, making it possible for multiple organ infections [70,71]. A number of recent studies have attempted to delineate the underlying mechanisms of COVID-19 related complications.

**Pulmonary complication**: Studies have shown a high proportion of pulmonary complications among survivors of severe COVID-19 infections who were on mechanical ventilation [72]. A three-month follow-up of COVID-19 survivors revealed abnormal chest computerized tomography (CT) in 42% of the patients [73]. Characteristic ARDS followed by pulmonary fibrosis and injury of alveolar epithelial cells caused by direct viral entry are a hallmark of COVID-19 infection [74]. The dysregulated levels of inflammatory cytokines including IL-6 and TNFα along with VEGF, cause a cytokine storm leading to fibrosis and further secondary bacterial infections [75]. The cytokine storm also plays a critical role in coagulation dysfunction leading to thrombocytopenia and stimulates megakaryocyte production leading to thrombocytosis [76].

**Cardiological complication**: Multiple studies have associated cardiac complications with COVID-19 infections. Initial studies from Wuhan reported heart failure among 23% of the COVID-19 patients [77]. While a study from the United States reported cardiac arrest in 14% of COVID-19 patients admitted to Intensive Care Units [78]. Several mechanisms have been postulated for cardiac injury post SARS-CoV-2 infection including direct viral entry with the presence of viral RNA in heart tissues, the inflammatory response often leading to necrosis and structural myocardial damage [79]. Studies have revealed increased cytokine levels indicative of myocardial infarction, endothelial dysfunction and plaque instability [80]. Increased cytokines, e.g., IL-1, IL-6 and TNFα due to COVID-19 may also raise catecholaminergic state causing arrhythmias [81].

**Neurological complication:** There is substantial evidence for neurological indisposition as COVID-19 sequelae. Studies have documented 33.6–80% incidence of neurological or psychiatric manifestations after 6–12 months in COVID-19 hospitalized patients [82,83]. The possible mechanisms of neuropathology due to COVID-19 include direct viral injury, microvascular thrombosis and systemic inflammation [84,85]. Along with the detection of viral RNA in the brain, neural tropism has been recently described [86]. The effects of accumulated memory T cells and low levels of neuro-inflammation may play a role in COVID-19 persistence. Another hallmark of COVID-19 is the loss of olfactory and gustatory senses mainly due to olfactory epithelium cell injury by the virus [87]. The entry of the virus into the host nervous system has been hypothesized through the neural-mucosal border present in the olfactory mucosa.

Renal complications post COVID-19 may be attributed to direct viral infection, acute kidney injury, fibrosis and systemic inflammation [88,89]. Additionally, the viral genome has also been detected in renal tissues during autopsies and biopsies [90,91].

Notably, many other COVID-19 related sequelae are being addressed and are coming into the picture with time, e.g., dermatologic, gastrointestinal, endocrine and genitourinary complications [92]. However, it is still not clear why a subset of SARS-CoV-2 infected individuals progress into disease severity while the resolution of disease and symptoms are observed in others. Few studies have illustrated that existing infection or previous exposure can substantially modulate the host immune response against SARS-CoV-2. In fact, it has been postulated that during COVID-19 related inflammation, existing pathogens concealed by SARS-CoV-2 may contribute to autoantibody generation which may continue to act after the resolution of acute phase infection leading to sequelae [93]. Reports of Guillain Barre syndrome, an autoimmune condition affecting the host neurological system, have been recorded in post COVID-19 patients [94,95].

## 6. COVID-19 Vaccines

Several COVID-19 vaccines are currently available and have been validated by the WHO. As of January 2022, eight vaccines have obtained the Emergency Use Listing (EUL) status (Table 4).

## 7. Effect of COVID-19 Vaccine on Host Immunity

The mRNA-based vaccines of COVID-19 induce maturation of CD4^+^ and CD8^+^ T lymphocyte and the vaccinated individuals tend to possess memory-based T-lymphocyte responses [107]. Individuals vaccinated with the second dose of mRNA-based vaccine show developed B-lymphocytes and high levels of IgM and IgG antibodies eight weeks post second dose of vaccine [108]. The vaccine induced memory B and T cells remain stable for up to 3–6 months post vaccination. Oxford AstraZeneca (AZD1222/ChAdOx1) and Sputnik V (gram-COVID-Vac-rAD26/rAD5) elicit receptor binding domain specific IgG and neutralizing antibody responses at >20 days after the first dose and further induction of responses takes place after second dose [108]. The S protein’s prefusion conformation has been targeted as the immunogen for vaccine formulation as it has significant epitopes against which neutralizing antibodies are made [108]. Delaying the interval between the first and the second dose of mRNA-based vaccines such as Oxford AstraZeneca (AZD1222/ChAdOx1) and Pfizer-BioNtech (BNT162b2) between 6 and 14 weeks results in higher neutralizing antibody levels than those of three-week intervals tested for clinical trials during vaccine licensing [98,109]. However, longer time intervals dampen the T-cell responses which might be attributed to the fact that longer time interval results in more S-specific T-cell response upon re-exposure to S-protein [110,111]. Recent reports suggest that COVID-19 vaccines boost the neutralizing antibody titers up to 10- to 45-fold higher in individuals with a history of infection compared to those without any past infection history after the first dose of vaccination [112,113]. However, the neutralizing antibody titers decline after six months of vaccination with a second dose as the vaccine induced plasma cells are short-lived compared to those acquired during natural infection [114,115].

**Waning effectiveness**: Prior to the outbreak of the Omicron variant, a few studies have evaluated the effectiveness of COVID-19 vaccines and observed that 7 months after a booster dose, the effectiveness declined for BNT162b2, mRNA-1273 and ChAdOx1 nCoV-19 plus an mRNA vaccine [116]. A few data also suggest that vaccine effectiveness relatively declined after 5 months of vaccination among older aged (above 55 years) and individuals with risk factors [117,118,119]. Therefore, administration of booster dose is assumed to reduce the risk of infection and restore the waning effect. A recent study in Israel evaluated the effect of fourth booster dose of BNT162b2 vaccine for effectiveness against the Omicron variant. The rate of infection considerably lowered for a short period while the elicited immunity did not wane for severe disease [120,121]. Another study found ‘viral load decrease effectiveness’ significantly declines after booster dose of BNT162b2 vaccine. It suggests a fast waning of effectiveness of vaccine’s booster dose possibly affecting community level infection [122]. With respect to the emerging variants of the virus, COVID-19 vaccines have been demonstrated to elicit protection against severe infection. However, variability in vaccine effectiveness is seen for protection against symptomatic infections. For example, data suggest that vaccine effectiveness against symptomatic infections caused by the Delta variant is lower than the wild type and Alpha variants [123,124]. The waning effectiveness post two and three doses of mRNA vaccines have been described during the Delta and Omicron variant pandemic wave further underscores the importance of additional doses to sustain protection against different variants of SARS-CoV-2 [125].

**Breakthrough infections**: As no vaccine is 100% effective, there are instances of infection even after receiving vaccination which are known as breakthrough infections. For COVID-19, there is a higher risk of breakthrough infection with Delta and Omicron variants but the risk of severe infections is low among individuals who received vaccine booster doses [126,127]. The first breakthrough infection with omicron was detected in fully vaccinated German individuals with mRNA vaccine and exhibited mild to moderate illness [128]. A recent study from the USA suggested that individuals with breakthrough infections showed a lower risk of fatality and post-acute sequelae as compared to unvaccinated individuals [129]. Moreover, it is indicative from the available data that the risk of breakthrough infections is higher among vaccinated immuno-compromised patients and individuals with existing comorbidities [128,130]. Recently, many retrospective cohort studies have been conducted among healthcare workers which show the association between COVID-19 vaccination and asymptomatic infections. Studies observed that individuals vaccinated with two doses of the BNT162b2 mRNA vaccine exhibited lower incidence rates of asymptomatic infection. However, these findings are limited by observational designs [131,132]. Similarly, a research letter from St. Jude’s Children’s Research Hospital reported that the vaccination greatly reduced the symptomatic and asymptomatic infections in employees compared with their unvaccinated contemporaries. However, an exact association with the asymptomatic infection remains unclear [133]. The impact of vaccine on the onset of asymptomatic infection depends on the efficacy of the specific vaccine. The reduction in the asymptomatic infection is also likely to be affected by the timing of the vaccine doses [100,134].

## 8. COVID-19 Vaccine-Related Complications

Millions of COVID-19 vaccine doses have already been administered worldwide and various health authorities that monitor vaccine safety, e.g., the WHO, CDC, United States Food and Drug Administration (USFDA) etc., as well as stakeholders, have repeatedly emphasized the safety of the vaccines in use. Although severe adverse events are extremely rare, anaphylactic events have been reported for around five cases per one million administered vaccine doses [135]. Anaphylaxis reported for COVID-19 vaccines includes cardiovascular, haematological, neurological and autoantibody related events.

**Cardiovascular**: As of June 2021, 1226 reports of myocarditis cases had been registered in the United States after the administration of more than 290 million doses of the COVID-19 mRNA vaccine [136]. Similar reports of myocarditis after vaccination have been reported from the Philippines, England and Israel [137]. Following the administration of two doses of mRNA-1273 (Moderna vaccine), BNT162b2 (Pfizer vaccine) and NVX-CoV2373 (Novavax vaccine), young males were reported with the occurrence of myocarditis and pericarditis [136,138,139]. Mostly mild myocarditis has been reported globally after a week of mRNA vaccine administration with speedy resolution of clinical symptoms [140].

**Hematological**: Around 0.0031% and 0.0045% of thrombotic events have been reported in the United Kingdom and Singapore, respectively, after administering mRNA vaccines [141]. Rare vaccine-related thrombosis and thrombocytopenia have been observed in five individuals of more than 1,30,000 cohort at around 7–10 days of first dose of the AstraZeneca vaccine (ChAdOx1 nCoV-19/AZD1222). The researchers proposed this adverse event as a spontaneous immune thrombotic thrombocytopenia induced by vaccine [142]. Of the approximately seven million doses of Ad26.COV2.S (Johnson & Johnson) vaccine administered, similar venous thrombosis with thrombocytopenia events have been recorded in 12 individuals by April 2021 after a single dose of vaccination. Fatal outcomes were recorded in 3 individuals [143]. The mRNA vaccines from Pfizer and Moderna also have shown the implications of triggering rare immune thrombocytopenia. A study observed 20 individuals with thrombocytopenia within 23 days of vaccination with two doses of the mRNA vaccines [144]. Although careful monitoring for symptoms post-vaccination is efficiently ongoing for such adverse events, vaccine regulatory bodies have reiterated the benefits of these COVID-19 vaccines in the population which outweigh rare risk events.

**Neurological**: Emerging reports of neurological symptoms as rare adverse events have been recorded post COVID-19 vaccination. A study from Hong Kong reported Bell’s palsy cases post Vaccination with the first dose of CoronaVac (Sinovac Biotech, Hong Kong, China) and BNT162b2 vaccines. They found 28 cases after CoronaVac vaccination of more than 4.5 million individuals while 16 cases were reported post BNT162b2 vaccination of 5.3 million individuals [145]. An increased risk of Bell’s palsy and Guillain Barre syndrome with 1.38 and 2.9 incidence rate ratio, respectively, was also observed after vaccination with the first dose of ChAdOx1 nCoV-19 [146]. After administration of more than 13 million doses of Ad26.COV2.S vaccine, around 130 cases of Guillain Barre syndrome cases have been reported after a single dose vaccination in the US [147]. Although the mechanism is still unclear, researchers have hypothesized the role of autoimmune response through stimulation of inactive autoreactive T cells or vaccine antigen mimicking host cells [148].

## 9. Future Implications and Recommendations

The global economic and public health impact of COVID-19 pandemic led to an unprecedented speed in collating data and information regarding all aspects of the disease. With the progress of COVID-19 pandemic, new evidence of characteristic underlying clinical conditions of the infection and risk factors for severity has come to the forefront. Although clinical features that can unfailingly distinguish COVID-19 from other respiratory viral diseases are not specific, literature evidence strongly indicates differences in quantitative and qualitative immune responses of SARS-CoV-2 infected patients which correlate with clinical presentations. There are yet unanswered gaps in understanding the pathogenesis of COVID-19 as to what drives the infection from asymptomatic, mild to severe and even leading to death in selected populations. Additionally, the consistent emergence of novel variants causing breakthrough infections worldwide suggests a decrease in the effectiveness of the available vaccines. Therefore, long term follow-up studies including large cohorts are essential to understand the effect of variants on vaccination as well as on the post COVID-19 sequelae complications.

In addition to the management of the clinical symptoms, vaccination against COVID-19 is the best shot at controlling the disease by preventing severity and deaths. However, reports of adverse events have created hesitancy in availing of vaccination in many parts of the world. It needs to be clearly and consistently communicated to the masses the outweighed benefits of vaccine as opposed to the few adverse events and thus encourage the population to make the positive choice of being vaccinated against SARS-CoV-2. Moreover, consistent monitoring and reporting of adverse events post vaccination should be mandated globally. The worldwide widespread vaccination programs can significantly improve the continued burden of the pandemic from the public healthcare system.

## Figures and Tables

**Figure 1 vaccines-10-01639-f001:**
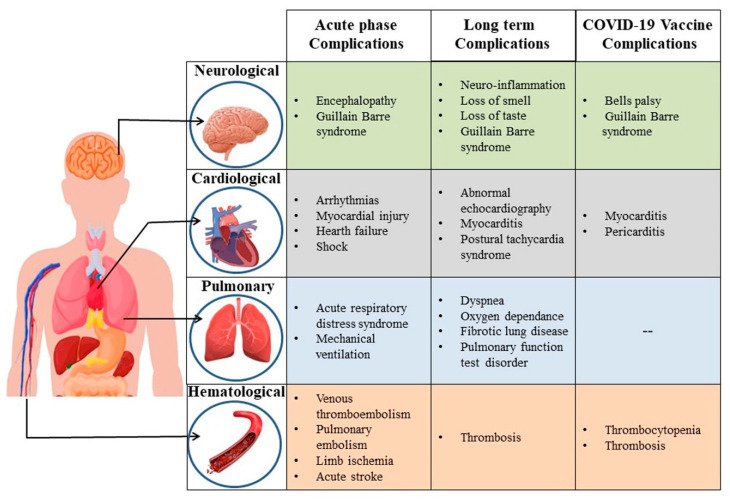
Schematic representation of complications in different organ system during acute and long-term sequelae of COVID-19 as well as adverse events post COVID-19 vaccination.

**Table 1 vaccines-10-01639-t001:** Variants of concern of the SARS-CoV-2 virus circulating globally.

Variants of Concern	Mutations with Potential Biological Effect	First Reported/Verified Circulation	First Detected	Transmissibility	Disease Severity	References
Alpha (B.1.1.7)	N501Y, 69–70del, P681H	UK/ 183 countries	Sep 2020	Increased compared with wild type	Increased compared with wild type	[17,18]
Beta (B.1.351)	N501Y, K417N, E484K	South Africa/ 121 countries	Sep 2020	Increased compared with wild type	Increased compared with wild type	[17,18]
Gamma (P.1)	E484K, N501Y, K417T	Brazil/ 74 countries	Oct 2020	Increased compared with wild type	Increased compared with wild type	[17,18]
Delta (B.1.617.2)	D614G, L452R, T478K, P681R	India/ 175 countries	Mar 2021	Increased compared with alpha	Increased compared with alpha	[17,18]
Omicron * (B.1.1.529)	R346K, L452X, F486V	South Africa/ Multiple countries	Oct 2021	Increased compared with delta	Decreased compared with delta	[17,18,19]

* Includes Omicron sub-lineages: BA.1, BA.2, BA.3 and descendant lineages.

**Table 2 vaccines-10-01639-t002:** Comorbidities associated with COVID-19 severity.

Comorbidities	References
Asthma	[54,55]
Age > 65 years	[56]
Cancer	[54,55]
Cardiovascular disease	[54,55]
Cerebrovascular disease	[54,55]
Chronic kidney disease	[54,55]
Chronic liver disease	[54,55]
Chronic lung disease	[54,55]
Cystic fibrosis	[54,55]
Diabetes mellitus (type 1 and 2)	[54,55]
Disabilities	[54,55]
Genetic haematological disorder	[54,55]
Immunodeficiencies	[54,55]
Infectious diseases, e.g., Tuberculosis and HIV	[54,55]
Mental health disorders	[54,55]
Neurological conditions	[54,55]
Obesity and overweight	[54,55]
Pregnancy	[54,55]
Smoking and substance abuse disorders	[54,55]
Organ transplant	[54,55]
Use of corticosteroids or other immunosuppressive medications	[54,55]

**Table 3 vaccines-10-01639-t003:** Genetic factors associated with COVID-19 severity.

Genes	Population	Association with COVID	References
*ABO*	Mostly Europeans	COVID-19 positivity	[58]
*HLA*	Chinese; Russian	Poor prognosis	[59,60]
*ACE2*	Multiple	Susceptibility	[61]
*IFITM3*	Spanish; Chinese	Severity	[62,63]
*OAS1*	Multiple	Severity	[64]

ABO: ABO gene; HLA: Human leukocyte antigen; ACE2: Angiotensin converting enzyme; IFITM3: Interferon induced transmembrane protein 3; OAS1: Oligoadenylate synthetase 1.

**Table 4 vaccines-10-01639-t004:** List of COVID-19 vaccines cleared for emergency use license by WHO.

Vaccine Name	Company (Manufacturing Locations)	Platform	Vaccine Efficacy	EUL Obtained	References
BNT162b2	Pfizer/BioNTech (USA; Europe)	mRNA	95%	Dec 2020	[96,97]
ChAdOx1 nCoV-19/Covishield/AZD1222	University of Oxford/Serum Institute of India/ AstraZeneca (UK; India)	Replication-incompetent adenoviral vector	70–76%	Feb 2021	[96,98,99]
mRNA-1273	Moderna (USA)	mRNA	94.1%	April 2021	[96,100]
WIV04 and HB02	Sinopharm (China)	Inactivated, whole-virus vaccine	73–78%	May 2021	[96,101]
CoronaVac	Sinovac (China)	Inactivated vaccine	70–83.5%	June 2021	[96,102,103]
Ad26.COV2.S	Janssen/Johnson & Johnson (Netherlands, Belgium)	Replication-incompetent adenovirus 26 vector	66.9%	March 2021	[100,103]
Covaxin/BBV152	Bharat Biotech (India)	Inactivated vaccine	78%	Nov 2021	[96,104]
NVX-CoV2373	Novavax (Europe; India)	Recombinant protein subunit	89–90.4%	Dec 2021	[96,105,106]

## Data Availability

Not applicable.

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
