# Peer review of "An Update on Complications Associated with SARS-CoV-2 Infection and COVID-19 Vaccination"

_vaccines, 2022, doi:10.3390/vaccines10101639_

Round 1

Reviewer 1 Report

It is timely review on SARS-CoV-2 infection and COVID-19 vaccination. In fact, authors seem to care about virus and its effect and Clinical spectrum. So the title could not be suitable. Moreover, some comments are showed as following:

1.      A review on whether there is relationship between asymptomatic infections and vaccination might be included or investigated.

2.      Information on complications of SARS-CoV-2 infection and COVID-19 vaccination is suggested to include epidemiological data although they are less.

Author Response

  1. A review on whether there is relationship between asymptomatic infections and vaccination might be included or investigated.

Ans: As per the Reviewer’s advice, relationship between COVID-19 asymptomatic infection and vaccination has been included in the manuscript (Breakthrough infection section).

  1. Information on complications of SARS-CoV-2 infection and COVID-19 vaccination is suggested to include epidemiological data although they are less.

Ans: As suggested, epidemiological data on complications of COVID-19 infection and vaccination have been incorporated in the manuscript.

Reviewer 2 Report

 The current review gives global information on COVID-19 infection as well as vaccination complications. It is an interesting and well-written review article that collects global recent information on the subject. The article contains an extended bibliography. The paper must be accepted however I think that a schematic color image will make the paper more attractive to readers.

Author Response

 The current review gives global information on COVID-19 infection as well as vaccination complications. It is an interesting and well-written review article that collects global recent information on the subject. The article contains an extended bibliography. The paper must be accepted however I think that a schematic color image will make the paper more attractive to readers.

Ans: We have included a figure representing acute, long-term sequelae of COVID-19 infection as well as adverse events post COVID-19 vaccination.

Reviewer 3 Report

This is the reviewer comment on manuscript entitled “An update on complications associated with SARS-CoV-2 infection and COVID-19 vaccination”. The manuscript ID is vaccines-1919242.

The manuscript submitted by Chowdhury et al. comprehensively reviews the clinical manifestations related to SARS-CoV-2 infection and COVID-19 vaccination. To achieve this goal, the authors discussed SARS-CoV-2 virus, viral pathogenesis, COVID-19 vaccines, host immune responses to both virus infection and vaccination, and most importantly, SARS-CoV-2 infection related clinical complications and COVID-19 vaccine related adverse effects. The manuscript is timely and likely to be of interest to readers of Vaccines. However, the authors should address the comments below:

1. The current organization of the manuscript is suggested to be re-organized as: 1). Introduction, 2). The virus and its effect, 3). Effect of SARS-CoV-2 infection in host immunity, 4). Clinical spectrum, 5). Complications during COVID-19, 6). COVID-19 vaccines, 7). Effect of COVID-19 vaccine in host immunity, 8). COVID-19 vaccine related complications, 9). Future implications and recommendations.

2. A major omission of this review manuscript is the clinical manifestations of SARS-COV-2 variants, eg. Alpha, Beta, Gamma, Delta, Omicron BA.1, Omicron BA.2, Omicron BA.5.

3. In Table 1 and Table 3, please provide the references in separate columns.

4. The vaccine related adverse effects should be addressed in respective to first, second, or booster doses.

5. The authors have not provided Figure 1 along with the manuscript while mentioned it in the text.

6. The authors need to improve the scientific writing in the introduction section, eg. “like wild fire”, “B-coronavirus”, “rigorous efforts of the scientists”, and the sentences “…after rigorous review passed an mRNA vaccine (BNT162) by Pfizer-BioNtech … of mass vaccination (7). As of June 2022, more than 11 billion doses of vaccine against COVID-19 have been administered worldwide” is confusing that Pfizer-BioNTech vaccine has been administered more than 11 billion doses.

7. Some narratives need fact-checking, eg. “as alternative antiviral treatment is still unavailable” which is not true because of therapeutic monoclonal antibodies, antiviral drugs etc. “Delaying the interval between the first and the second dose of mRNA-based vaccine like Oxford AstraZeneca (AZD1222/ChAdOx1)”.

8. The authors need to check the grammars throughout the manuscript, eg. “clinical manifestations of COVID-19 ranges”, “…COVID-19 into five categorizes viz”.

Author Response

 (Reviewer 3)

Comments and Suggestions for Authors

This is the reviewer comment on manuscript entitled “An update on complications associated with SARS-CoV-2 infection and COVID-19 vaccination”. The manuscript ID is vaccines-1919242.

The manuscript submitted by Chowdhury et al. comprehensively reviews the clinical manifestations related to SARS-CoV-2 infection and COVID-19 vaccination. To achieve this goal, the authors discussed SARS-CoV-2 virus, viral pathogenesis, COVID-19 vaccines, host immune responses to both virus infection and vaccination, and most importantly, SARS-CoV-2 infection related clinical complications and COVID-19 vaccine related adverse effects. The manuscript is timely and likely to be of interest to readers of Vaccines. However, the authors should address the comments below:

  1. The current organization of the manuscript is suggested to be re-organized as: 1). Introduction, 2). The virus and its effect, 3). Effect of SARS-CoV-2 infection in host immunity, 4). Clinical spectrum, 5). Complications during COVID-19, 6). COVID-19 vaccines, 7). Effect of COVID-19 vaccine in host immunity, 8). COVID-19 vaccine related complications, 9). Future implications and recommendations.

Ans: The manuscript has been rearranged as per the reviewer’s suggestion.

  1. A major omission of this review manuscript is the clinical manifestations of SARS-COV-2 variants, eg. Alpha, Beta, Gamma, Delta, Omicron BA.1, Omicron BA.2, Omicron BA.5.

Ans: We thank the reviewer for the valuable comment. Details on transmissibility, disease severity, etc. of the SARS-CoV-2 variants of concern have been incorporated in the manuscript.

  1. In Table 1 and Table 3, please provide the references in separate columns.

Ans: As advised, the references have been incorporated in separate columns for both Table 1 and 3.

  1. The vaccine related adverse effects should be addressed in respective to first, second, or booster doses.

Ans: COVID-19 vaccine related complications have been addressed with respect to their reported vaccination doses in the corrected manuscript.

  1. The authors have not provided Figure 1 along with the manuscript while mentioned it in the text.

 Ans: We have duly incorporated the figure.

  1. The authors need to improve the scientific writing in the introduction section, eg. “like wild fire”, “B-coronavirus”, “rigorous efforts of the scientists”, and the sentences “…after rigorous review passed an mRNA vaccine (BNT162) by Pfizer-BioNtech … of mass vaccination (7). As of June 2022, more than 11 billion doses of vaccine against COVID-19 have been administered worldwide” is confusing that Pfizer-BioNTech vaccine has been administered more than 11 billion doses.

 Ans: The introduction section of the manuscript has been thoroughly corrected as per the reviewer’s advice.

  1. Some narratives need fact-checking, eg. “as alternative antiviral treatment is still unavailable” which is not true because of therapeutic monoclonal antibodies, antiviral drugs etc. “Delaying the interval between the first and the second dose of mRNA-based vaccine like Oxford AstraZeneca (AZD1222/ChAdOx1)”.

Ans: “as alternative antiviral treatment is still unavailable”: We have deleted the mentioned statement from the manuscript.

“Delaying the interval between the first and the second dose of mRNA-based vaccine like Oxford AstraZeneca (AZD1222/ChAdOx1)”: Delaying the interval between the first and the second dose of mRNA-based vaccines like Oxford AstraZeneca (AZD1222/ChAdOx1) and Pfizer-BioNtech (BNT162b2) between 6-14 weeks, results in high neutralizing antibody levels than those of three-week intervals tested for clinical trials during vaccine licensing. However, the delay duration must take into consideration some crucial information like the extent of protection provided in a single dose, emergence of highly virulent COVID-19 variants and the expected vaccine supply pipeline. This statement as been supported by a number of references (Voysey 2021; Silva 2021; Parry 2021; Payne 2021).

  1. The authors need to check the grammars throughout the manuscript, eg. “clinical manifestations of COVID-19 ranges”, “…COVID-19 into five categorizes viz”.

Ans: We have carefully checked the grammar for the entire manuscript and have made the necessary corrections.

Round 2

Reviewer 1 Report

Authors have addessed the comments, and the manuscript has been revised.

Reviewer 3 Report

The authors have addressed all the comments in the revision. No further comments are raised and the acceptance is suggested for the current version of manuscript.